# Resistance Mechanisms in Pediatric B-Cell Acute Lymphoblastic Leukemia

**DOI:** 10.3390/ijms23063067

**Published:** 2022-03-12

**Authors:** Krzysztof Jędraszek, Marta Malczewska, Karolina Parysek-Wójcik, Monika Lejman

**Affiliations:** 1Student Scientific Society of Laboratory of Genetic Diagnostics, Medical University of Lublin, 20-093 Lublin, Poland; krzysztofjedraszek13@gmail.com; 2Department of Pediatric Hematology, Oncology and Transplantology, Medical University of Lublin, 20-093 Lublin, Poland; martmalczewska@gmail.com (M.M.); parysek.kar@gmail.com (K.P.-W.); 3Laboratory of Genetic Diagnostics, Medical University of Lublin, 20-093 Lublin, Poland

**Keywords:** acute lymphoblastic leukemia, resistance, treatment resistance

## Abstract

Despite the rapid development of medicine, even nowadays, acute lymphoblastic leukemia (ALL) is still a problem for pediatric clinicians. Modern medicine has reached a limit of curability even though the recovery rate exceeds 90%. Relapse occurs in around 20% of treated patients and, regrettably, 10% of diagnosed ALL patients are still incurable. In this article, we would like to focus on the treatment resistance and disease relapse of patients with B-cell leukemia in the context of prognostic factors of ALL. We demonstrate the mechanisms of the resistance to steroid therapy and Tyrosine Kinase Inhibitors and assess the impact of genetic factors on the treatment resistance, especially *TCF3::HLF* translocation. We compare therapeutic protocols and decipher how cancer cells become resistant to innovative treatments—including CAR-T-cell therapies and monoclonal antibodies. The comparisons made in our article help to bring closer the main factors of resistance in hematologic malignancies in the context of ALL.

## 1. Introduction

Acute lymphoblastic leukemia (ALL) is the most common blood cancer in the pediatric population. According to the American Cancer Society, in the US in 2021, there were 5690 new cases and 1580 deaths due to ALL in adults and children [1]. The rate of new cases of ALL in 2019 was 1.8 per 100,000 men and women per year, and the death rate was 0.4 for every 100,000 men and women over the year [2]. Similarly, in Europe, the rate of new cases is 40 per million Europeans. Among Asians, the rate of new cases is 1.6 for every 100,100 people [3,4]. Based on immunophenotype, pediatric ALL may be of B- (B-ALL) or T-lymphoid (T-ALL) lineage, accounting, respectively, for 85% and 15% of all cases. Patient age and WBC diagnosis were long-standing features incorporated into risk-stratification algorithms for patients with B-ALL. Regardless of the immunophenotype, minimal residual disease (MRD) assessment at designated time points became the most significant predictor of outcome. High-throughput profiling of genetic alterations has allowed us to better understand the leukemogenesis and therapeutic response and identify risk-associated ALL genetic subtypes. The above-mentioned prognostic factors are utilized to risk-stratify patients and tailor therapy at the time of diagnosis. Currently, in B-ALL, there are over 30 subtypes described by patterns of genetic alterations, including aneuploidy, chromosomal rearrangements, sequence mutations, and, typically, distinct gene expression profiles. The identification of well-known risk-associated genetic subtypes, such as the rearrangement of *ETV6/RUNX1*, *BCR/ABL1*, *TCF3*, *KMT2A*, hyperdiploid or hypodiploid karyotype, or status of *IKZF1*, is a crucial part of every diagnosis and therapy [5,6]. Newly described genetic subtypes, such as *MEF2D*, *DUX4*, *NUTM1*, *ZNF384*, *PAX5*, *Ph-like ALL*, and *ETV6/RUNX1*, still require further studies to determine their prognoses and outcomes. Nevertheless, many of these findings are of direct clinical significance in leukemia diagnoses and have great potential to improve the prognosis/survivability of patients with B-ALL. 

Currently, the recovery rate exceeds 90% in children; however, the relapse rate of treated ALL patients is roughly 20%, and the remaining 10% of patients are still incurable. Notably, conventional chemotherapy has reached its limit in terms of its curability of pediatric ALL patients [7,8]. Despite the fast advancement of therapy and new medicine development, ALL remains a persistent concern among the pediatric population, where resistance to treatment widely occurs. This phenomenon can be observed, i.e., in microbiology, diabetology, and oncology. The epigenetic and genetic changes determine variability in leukemia cell activities. Acquiring resistance occurs because of changes in DNA-proliferating cells, where new random factors can cause the expression or inhibition of genes, both before and after implementing therapy. Ways to overcome resistance include the use of non-standard chemotherapy, immunotherapy, allo-HSCT, or CAR-T cell therapy [5,9]. The high percentage of relapses and resistance to treatment continues to be discouraging. It is crucial to investigate cancer cells’ resistance to currently used therapies. Achieving a new understanding of ALL biology is an unmet need to expand therapeutic options and overcome treatment resistance. 

In this review, we discuss the different mechanisms of blast cell resistance to the types of therapy used in B-cell ALL. We present the genetic abnormalities responsible for resistance to steroids and cytostatics, the interdependence between genes found in sensitive and resistant cells, and their effects on the metabolism environment of the cell. 

## 2. Results

### 2.1. Standard Therapeutic Protocols Used in Developed Countries 

Worldwide, different treatment protocols are used to treat childhood Acute Lymphoblastic Leukemia (Table 1). Compared and selected treatment protocols are used in developed countries. In Europe, ALL is treated according to the International Collaborative Treatment Protocol for Children and Adolescents with Acute Lymphoblastic Leukemia—AIEOP BFM ALL 2017. To cure ALL, the Children Oncology Group Protocol (COG-AALL) is applied in the US. The United Kingdom National Randomized Trial For Children and Young Adults with Acute Lymphoblastic Leukemia and Lymphoma 2011 (UK ALL 2011) is used on patients treated in Great Britain. Childhood ALL in China is cured with the Chinese Children Cancer Group Study—ALL 2015 (CCCG-ALL-2015). Japan uses the Japan Association of Childhood Leukemia Study (JACLS) to cure ALL. Despite the differences in the above-mentioned protocols, the general treatment is based on conventional chemotherapy that can be divided into specific stages. This includes the induction of remission, consolidation, and the maintenance of remission. The core of treatment is steroid therapy, accompanied by cytostatics. The most commonly used drugs are Vincristine (VCR), Daunorubicin (DNR), Methotrexate (MTX), Doxorubicin (DOXO), Cytarabine (ARA-C), Cyclophosphamide (CMP), Thioguanine (TG), and 6-mercaptopurine (6-MP).

### 2.2. Genetic and Metabolic Mechanism of Resistance for Glucocorticoids and Cytostatics

Glucocorticoids (GCs)—prednisone and dexamethasone—remain an essential part of the global treatment of acute lymphoblastic leukemia. GCs are administered in a constant dose for about 4 weeks. In most protocols, response to steroids is assessed after 7 days of therapy. A good response (GR) is defined as a decrease in the number of blast cells in peripheral blood (a peripheral blast count less than 1000/mm^3^). A poor response (PR) to steroids (peripheral blasts ≥ 1000/mm^3^) is defined as a failure to achieve effective cytoreduction, and it is strongly associated with the risk of relapse [18]. Although a considerable majority of patients with acute lymphoblastic leukemia react efficiently to GC therabpy in the early stages, extended treatment can result in steroid resistance [19]. Prednisone and dexamethasone are cortisol analogs that interact with the glucocorticoid receptor NR3C1. Activation of these receptors is responsible for many reactions in the human body, such as decreased inflammation, increased blood sugar level, increased food intake, reduced bone formation, and the increased breakdown of proteins [20]. *NR3C1* (locus 5q31.3) is a gene that encodes the human glucocorticoid receptor (hGR), a ligand-dependent transcription factor. Glucocorticoid-responsive genes are activated by binding directly to promoter regions of glucocorticoid response elements (GREs) or by modulating the transcriptional activity of other factors via protein–protein interactions. hGR positively or negatively regulates almost 20% of genes that are expressed in human leukocytes [21,22]. Mutations or polymorphisms in the hGR gene impair one or more of the molecular mechanisms of hGRalpha action and, as a final consequence, alter tissue sensitivity to glucocorticoids. The GCs resistance was associated with mutations at the level of the glucocorticoid receptor (some of which were newly identified and previously not associated with GC resistance, such as A484D, P515H, L756N, Y663H, L680P, and R714W0 [23]. The survival probabilities in children with ALL were associated with homozygosity of the G allele of the *NR3C1* BcII polymorphism, presenting a worse progression and prognosis of the disease [24]. Three other *NR3C1* SNP polymorphisms, −627A/G, intron 2 +646C/G and 9bT/C, were associated with dismal childhood cALL outcomes with reduced event-free and overall survival [25]. In 2018, Jing and colleagues proved that the cause of inadequate response to steroids lies in the disordered transmission of signals via the NR3C1 receptor. In their research, they analyzed the lymphoid-specific chromatin regions that bind with the NR3C1 receptor, both in GC-sensitive and GC-resistance cells. They documented changes in acetylation in one of the histones—H3K27—that leads to the consolidation of chromatin and, as a result, to the silencing of genes located there. Further analysis showed that the fragment of chromatin that has been consolidated in resistant cells corresponded to the so-called intron enhancer for the *BCL2L11* gene. The intron enhancer affects gene promoters with the use of chromatin loops, cohesin complexes, and the CCCTC binding factor CTCF. More CTCF activity with the *BCL2L11* gene was detected in steroid-sensitive cells. In contrast, blast cells’ resistance to steroids has shown to be due to a lack of the CTCF factor, resulting in a loss of upregulation of the *BCL2L11* gene [26]. *BCL2L11* (locus 2q13) is a BCL2-like 11 apoptosis facilitator (HGNC Alias symbol is BIM). BCL2L11/BIM is a BH3-only protein from the Bcl-2 family. Bcl-2 family members are the main regulators of programmed cell death via the mitochondrial (intrinsic) apoptotic pathway [27]. Furthermore, apart from the availability of chromatin, it has been postulated that chemotherapy of ALL induces actual drug-resistance mutations. A genome of 103 patients with relapsed ALL was examined and 12 genes were identified, 11 of which are associated with leukemia relapse. This applies to mutations in the following genes: corticosteroid receptors *NR3C1* (locus 5q31.3) [28] and *NR3C2* (locus 4q31.23) [29]; epigenetic regulators *CREBBP* (locus 16p13.3) [30] and *WHSC1* (locus 4p16.3) [31]; nucleotide-metabolism enzymes *NT5C2* (locus 10q24.32-q24.33) [32], *PRPS1* (locus Xq22.3), and *PRPS2* (locus Xp22.2) [33]; DNA-mismatch repair genes *MSH2* (locus 2p21-p16.3), *MSH6* (locus 2p16.3) [34], and *PMS2* (locus 7p22.1) [35]; and the tumor-suppressor gene *TP53* (locus 17p13.1) [36].

In Mullighan et al., the analysis of an expanded cohort of 341 ALL patients revealed that nearly 20% of relapsed cases had *CREBBP* sequence or deletion mutations [37]. The *CREBBP* gene encodes the transcriptional coactivator and histone acetyltransferase CREB-binding protein [38]. The most common mutation in the *CREBBP* gene is missense mutations that result in amino acid substitutions in the histone acetyltransferase (HAT) domain. Mutations in the HAT domain result in the decreased expression of CREB target genes, impaired cell proliferation, and disrupted expression of glucocorticoid-receptor-responsive genes. After examining cell lines with the presence of the *CREBBP* mutation, it was found that most of them were resistant to steroids [37]. According to the scientists, HDAC inhibitor therapy may be beneficial in steroid-resistant B-ALL with *CREBBP* mutation. Histone deacetylases (HDACs) are enzymes that regulate chromatin shape and function by removing acetyl residues from core histones, keeping chromatin in a transcriptionally quiet state, which makes them important in anticancer therapy. However, identifying the exact pathways by which these inhibitors induce cancer cell death still remains challenging [39]. 

A prognostic factor for the risk of recurrence, in addition to a poor response to steroids, is the response to subsequent drugs introduced during the protocol. Resistance to Methotrexate (MTX) is a common cause of relapse in children. MTX is infused during consolidation as a high dose of MTX (HD-MTX), administered intrathecally during the whole treatment and orally during the maintenance of remission [11]. 

MTX is a folate analog that inhibits enzymes involved in purine synthesis, necessary for the survival of rapidly replicating cells. Enzymes especially dependent on the MTX effect especially are dihydrofolate reductase (DHFR) and thymidylate synthase (TS) [40]. The *DHFR* gene (locus 5q14.1) encodes dihydrofolate reductase, an enzyme that converts dihydrofolate to tetrahydrofolate [41,42]. The *TYMS* gene (locus 18p11.32) expresses thymidylate synthase (TS), which is involved in the production of thymidylate [43]. Those enzymes catalyze the reactions necessary for purine synthesis. TS also plays an essential role in regulating cell metabolism [44]. The enzymes DHFR and TS are inhibited by MTX, which is a tightly binding but reversible inhibitor. As a result of the blockage of this route, dihydrofolate accumulates, and thymidylate synthase and purine production are inhibited [45]. If MTX cannot affect DHFR and TS and reduce their levels, purine synthesis will not be stopped, which will lead to the continuous proliferation of cells. Such a mechanism (opposite to MTX activity) will have an effect on the drug resistance of cells [46]. 

Methotrexate is carried by the reduced folate transporters—RFC or SLC19A1. Following this, a polyglutamylation reaction occurs inside the leukemic cell, which contributes to the inhibition of MTX. The tumor cell also leads to an increased MTX efflux. However, polyglutamylation of MTX is prevented by folylpolyglutamate hydrolase (FPGH) and folylpolyglutamate synthetase (FPGS). *FPGS* (locus 9q34.11) codes the enzyme, which is required for folate homeostasis and the survival of proliferating cells because it plays a role in building and maintaining both cytosolic and mitochondrial folylpolyglutamate concentrations [47]. FPGS is also a semi-finished product of the final FPGH enzyme. In the cytoplasm, MTX undergoes polyglutamylation—a conversion mediated by FPGS, resulting in the formation of a long chain of MTX. Studies were conducted to measure the level of MTX. In A. Wojtuszkiewicz et al.’s study, clinical data were collected from 235 children with ALL who were classified as MTX-resistant. The measurement of MTX polyglutamate concentrations in leukemic cells, mRNA expression of enzymes involved in MTX metabolism (FPGS, FPGH, RFC, DHFR, and TS), MTX sensitivity, and FPGS activity were all part of the study. The outcome of the study was that higher MTX long-chain polyglutamate accumulation is closely correlated with improved overall and event-free survival [48].

The sensitivity of the cell to MTX was analyzed concerning different genetic anomalies. Leukemic cells that present the hyperdiploid karyotype have an increased expression of the *RFC* gene locus. The *RFC1* gene (locus 4p14) encodes the major subunit of replication factor C and accessory protein for DNA polymerase, which is required for the coordinated synthesis of both strands of DNA (during its replication or repair) [49]. Furthermore, Uchiumi et al. hypothesized that *RFC* plays a role in telomere stability or rotation [50]. C. Kinahan et al. conclude that abnormalities in *RFC* expression lead to impaired cell survival. In contrast, increased expression of the *RFC* gene leads to the resistance of factors that promote cell death. Overexpression of *RFC* is also associated with the presence of an extra 21 chromosome. Cells that have mutations (at nucleotide position 227 where Guanine changes to Adenosine; a Cytosine to Thymine mutation at nucleotide position 352) or less expression of the *RFC* gene show resistance to MTX [48,51]. ALL patients with *ETV6/RUNX1* fusion or *KMT2A* (locus 11q23.3) rearrangement present resistance for MTX. In some cases, patients with *ETV6/RUNX1* have a higher level of *ABCG2* expression (locus 4q22.1)—a transporter that pumps MTX out of the cell [52,53].

Esti Liani and colleagues summed up a variety of mechanisms of MTX resistance. These mechanisms include: (a) single amino acid changes in DHFR, increased DHFR overexpression, or decreased affinity of MTX for DHFR; (b) defective membrane transfers due to changes in the *RFC*’s quality or quantity; (c) the presence of MRP1 and MRP3 proteins, so-called multidrug resistance proteins (MRPs), enhancing antifolate efflux [54]; (d) reduced antifolate polyglutamylation due to changes in *FPGS* activity, either quantitatively or qualitatively; (e) higher *FPGS* activity or a decrease in folate efflux transport function, resulted in increased cellular folate pools [55].

MTX resistance is not the only type that occurs in ALL patients. Genetic abnormalities associated with asparaginase resistance were observed in leukemic cells. Asparaginase has been used to cure ALL since 1967 [56]. Asparaginase produces its anti-tumor effects by converting the protein asparagine to aspartic acid. The *ASNS* gene (locus 7q21.3) is responsible for encoding asparagine synthetase, an enzyme that catalyzes the transfer of ammonia from glutamine to aspartic acid, resulting in the formation of asparagine [57]. Blast cells show a low level of aspartate synthetase (ASNS) and show no up-regulation after asparaginase exposure. The inability to synthesize one’s own asparagine and the low level of asparagine in the extraventricular fluid leads to the death of the cancer cell. Accessible medical asparaginase is an enzyme isolated from *Escherichia coli* and *Erwinia chrysanthemi*. FDA-approved drugs are Native *Escherichia coli* asparaginase (Elspar), the Pegylated form of native *Escherichia coli* asparaginase (Oncaspar), and *Erwinia chrysanthemi* asparaginase (Erwinse) [58]. Studies in mice, completed by B. Horowitz et al., have shown that resistant cells have a high level of ASNS, and sensitive cells showed little or no protein at all [59]. These reports were confirmed in patients with ALL. High expression of the *ASNS* gene was also found in patients with t(12;21) translocation. In t(12;21)+ blast cells, the ASNS concentration was significantly lower [60]. Many scientific studies have shown an increase in ASNS in blast cells after drug exposure. In those cells, up-regulation has been found, which can last up to 6 weeks after drug administration. Unfortunately, this regulation has been documented not only in asparaginase-resistant but also in asparaginase-sensitive cells [61,62,63]. The mechanisms of resistance, which are not fully understood, are substrate changes for ASNS. In asparaginase-resistant cells, the increased activity of glutamate synthetase (GD) and increased uptake of glutamine from the extracellular space have been demonstrated [56]. The use of Asparaginase is limited by its side effects, most notably severe allergic reaction, acute pancreatitis, and coagulation disorders [64]. Table 2 summarizes the genes responsible for steroid and cytostatic resistance discussed in this chapter.

### 2.3. Early Relapse as an Effect of Mutations in Genes Involved in the Metabolism of Purine Analogs

Early relapse is often associated with resistance to 6-mercaptopurine (6-MP), the mainstay of maintenance remission in most protocols. Resistance to 6-MP may be the result of gain-of-function mutations in *NT5C2*, mutations in *the PRPS1* gene, and loss of the *MSH6* gene. *NT5C2* (locus 10q24.32-q24.33) encodes an enzyme that catalyzes the dephosphorylation of neosine, guanosine, and xanthine monophosphates. *NT5C2* acts in opposition to the activity of nucleoside kinases by regulating the number of purine nucleotides and their efflux from the cell. The dephosphorylation reaction is enhanced, and the concentration of nucleotide monophosphates produced by 6-MP—methylthio-IMP and deoxythioguanosine triphosphate—is reduced. These compounds mediate the anti-tumor effect of 6-MP. Mutations in the *NT5C2* gene involving upregulation are found in 3–10% of patients with relapsed B-ALL [65]. In their work, Ch. Dieck and A. Ferrando discuss the mechanism of relapse, with up to 90% of patients showing the *NT5C2 R367Q* mutation. Leukemias with *NT5C2* mutations are chemoresistant to 6-mercaptopurine, yet show impaired proliferation and self-renewal [65]. There are three common variants of mutations in this gene. *NT5C2* class I mutations (*NT5C2 K359Q* and *L375F*) drive the helix A segment into the active helix configuration, resulting in high levels of nucleotidase activity. Class II *NT5C2* mutant proteins, on the other hand, have enhanced nucleotidase activity under baseline circumstances but still respond dynamically to allosteric stimulation. Positively charged residues in the intermonomeric pocket are disrupted by Class II mutations. Finally, *NT5C2* class III alterations are characterized by a C-terminal truncating allele (*NT5C2 Q523**), a mutant whose nucleotidase activity under baseline circumstances is comparable to that of the wild-type *NT5C2* protein but exhibits a significantly enhanced response to allosteric activation. The *Q523** mutation, which removes the C-terminal tail of *NT5C2*, generates a more open conformation and improves sensitivity to allosteric activators [66]. Although class I mutations may require direct targeting of the catalytic center, class II and class III *NT5C2* mutant proteins retain sensitivity to allosteric regulation, implying that small molecule inhibitors targeting the allosteric site may effectively abolish their ability to confer 6-MP resistance. This mechanism can be overcome by increasing the dose of 6-MP; of course, it is limited by the toxicity of the drug (myelosuppression, liver damage) [65].

Another study, conducted by Lui et al., indicates an increased risk of ALL relapse in patients with a mutation in the phosphoribosyl pyrophosphate synthetase 1 gene (*PRPS*). *PRPS* encodes enzymes that catalyze the phosphoribosylation of ribose 5-phosphate to 5-phosphoribosyl-1-pyrophosphate. This reaction is necessary for purine and pyrimidine biosynthesis. Three *PRSP* genes have been identified: widely expressed *PRSP1* (locus Xq22.2), *PRSP2* (locus Xq22.3), and *PRSP3* (transcripted only in testis) [33]. The study involved 348 patients, and a mutation of this gene occurred in 6.7% of them. Seventeen different nonsynonymous substitutions were found, and the most frequent one was *A190T.* This substitution leads to gain-of-function mutations, in effect leading to reduced activity of enzymes that metabolize purine analogs, impacting the efficiency of 6-MP and 6-thioguanine (6-TG), which are important in ALL therapy [67,68]. 6-TG is derived from 6-MP and impedes replication and transcription processes, inducing cytotoxicity, by intercalating within the DNA, resulting in DNA damage and target cell apoptosis [69]. 

Specific hemizygous deletion on chromosome 2p16.3 involving the *MSH6* gene was identified in about 4–10% of the examined patients with relapse ALL [70]. *MSH6* is a component of the mismatch repair (MMR) system and encodes protein products that play a role in strand-specific DNA-replication-error repairs. Mismatched DNA provokes the exchange from ADP to ATP, resulting in converting the MSH2-MSH6 complex to repair machinery sliding along the DNA backbone [71]. In their work, Evensen et al. demonstrated that lower expression of *MSH6* was associated with increased resistance, not only to 6-mercaptopurine, but also for prednisone. The mechanism of thiopurines is based on the insertion of thioguanine (TGN) instead of cytosine (Cyt). The MMR complex recognizes the mismatch and attempts to repair the DNA. It is not certain if DNA damage (preceded by attempts to repair it) or mismatch recognition alone leads to cell apoptosis. The loss of *MSH6* causes a lack of cell apoptosis, which leads to an increase in blast survival, despite the presence of thiopurines [70]. 

Another study, conducted on a group of 125 relapsed patients, showed that in 12% of cases of cells mutations, the *SETD2* gene was detected/involved. The Set-Domain-Containing Protein 2 (*SETD2*) (locus 3p21.31) is the primary methyltransferase catalyzing H3K36 trimethylation, which is related to DNA repair [72]. In human cancer, *SETD2* has been shown to play a tumor-suppressor role [73]. Mar et al. identified 24 mutations, the most common of which were loss-of-function, frameshift, and nonsense mutations, which lead to a decrease in gene expression. This results in a lack of tumor suppression. These researches also demonstrated the coexistence of *SETD2* mutations in the ALL subtypes with *KMT2A* rearranged (22%) and *ETV6-RUNX1* (13%) [74]. Table 3 summarizes the mutations of genes associated with the metabolism of purine analogues, the effect of these mutations on pathology and treatment.

### 2.4. Hyperdiploidy—Better Prognosis?

In the World Health Organization’s classification of hematopoietic and lymphoid tissue tumors, hyperdiploidy (HD) in B-ALL is defined by the presence of above 47 chromosomes in a somatic cell and is classified as a distinct subtype of B-ALL [75]. One subtype of HD is high hyperdiploidy (karyotypes containing modal chromosome number of 51–67 chromosomes), occurring in up to 30% of all patients with B-ALL. It is characterized by a nonrandom distribution of chromosome gains 21, X, 14, 6, 18, 4, 17, and 10 in the karyotype [75,76]. These patients have a good prognosis with survival rates exceeding 90%. In a Children’s Oncology Group referral study, 56 patients with standard-risk B-ALL and undetectable MRD (at day 29) achieved an exceptional 5-year event-free survival rate of 98.1 percent and an overall survival rate of 100 percent [77]. Despite a good prognosis, a significant percentage of this group—even up to 25%—have relapses [78]. An increased risk of recurrence is associated with various mutations that are present in cells with hyperdiploidy. Smith et al. have conducted a study on 57 patients with HHD-ALL. Most HHD-ALL patients had somatic mutations in at least one RTK/Ras/MAPK signaling pathway gene and in the epigenetic regulation gene CREBBP [79]. The Ras/Raf/MAPK pathway is responsible for transmitting signals from the extracellular environment to the cell nucleus, where genes responsible for cell cycle regulation, wound healing, and tissue repair are activated. As a result, overactive Ras/Raf/MAPK signaling affects a number of cellular activities that are critical for carcinogenesis [80,81]. Genes associated with the Ras/Raf/MAPK pathway are *KRAS*, *NRAS*, *FLT3*, and *PTPN11*. The *KRAS* (locus 1p13.2) protooncogene uses protein dynamics to operate as a molecular on/off switch. After allosteric activation, it recruits and activates proteins necessary for growth factor proliferation, as well as other cell-signaling receptors such as c-Raf and PI 3-kinase [82]. Next, the *NRAS* protooncogene (locus 1p13.2) provides instructions for the formation of a protein called N-Ras, which is largely involved in the regulation of cell division. These proteins release signals that instruct cells to grow, divide (proliferate), mature, and differentiate [83]. Oshima et al. confirmed the presence of mutations that activate the Ras/Raf/MAPK pathway in leukemic cells in patients with relapse. They performed whole-genome sequence analysis on 55 patients, 22 with B-ALL. Somatic mutations were described in 17 DNA samples of recurrent leukemia. Among these 17 mutations, already known genes responsible for B-ALL relapse were identified, including (responsible for steroid resistance) *CREBBP*, *KRAS*, *NRAS*, and Janus kinase 2 and 3 (*JAK2*, *JAK3*). They then isolated leukemia cell lines with *KRAS* mutations. These cells showed increased resistance to MTX and increased sensitivity to vincristine [84]. MTX resistance was directly proportional to *KRAS* mutation expression and Ras/Raf/MAPK activation. Another interesting identification was a mutation in the *FLT3* gene. In HHD instances, activating point mutations in the *FLT3* gene have been found in 20–25 percent of cases, compared to 9–14 percent in pediatric ALLs in general [85]. *FLT3* (locus 13q12.2) gene codes for the protein fms-like tyrosine kinase 3 (FLT3), which belongs to the receptor tyrosine kinase family (RTKs). The FLT3 protein also activates signaling pathways that control a variety of important cellular processes, particularly in early blood cells known as hematopoietic progenitor cells [86]. In Smith’s previously mentioned study, the FLT3 mutation co-occurred with a mutation in the *DOT1L* gene [79]. *DOT1L* (locus 19p13.3) regulates gene transcription, development, cell cycle progression, somatic reprogramming, and DNA damage repair [87]. Previously, this mutation was identified in an HD-ALL patient, indicating that it is typical for this subgroup. Additionally, *DOT1L* has been associated with the development of ALL with MLL (mixed-lineage leukemia) rearrangements [88]. The *PTPN11* gene, which also activates the Ras/Raf/MAPK pathway (locus 12q24.13), produces a protein that belongs to the protein tyrosine phosphatase (PTP) family responsible for cellular processes regulation [89,90]. Subsequent studies conducted by Junichi Ueyama et al. also confirmed the presence of *CREBBP* mutation and the activity of the RTK/RAS pathway in relapsed high-hyperdiploid ALL [79,91]. Another aspect of HHD abnormalities was found by Adam J. de Smith et al. They found that *GAB2*, a suspected new ALL predisposition gene, was another upstream stimulator of *RAS* signaling. GAB2 belongs to the GRB2-associated binder adaptor/scaffolding protein family and is a key binding partner of the SHP2 protein encoded by *PTPN11*. The Ras/Raf/MAPK and PI3K/AKT signaling pathways are activated when GAB2 binds to SHP2 and the p85 regulatory subunit of PI3K, respectively [92]. Finally, an interesting discovery was the identification of a somatic mutation in the *ROS1* gene (locus 6q22.1), which encodes a protein tyrosine kinase receptor [93]. Until now, this gene has been associated with lung cancer and glioblastoma. The ROS1 protein mediates phosphorylation, thereby increasing activation of the SHP2 protein (encoded by PTPN11). *ROS1* activation may promote cancer development. Given that ROS1 inhibitors—crizotinib or foretinib—are available for medical treatment, this discovery may contribute to another therapeutic line. However, further research on this subject needs to be completed [79]. Table 4 showsthe genes and their mutations that co-occur with hyperdiplody and are responsible for treatment resistance in this group of patients.

### 2.5. Translocations Responsible for Worse Prognosis

#### 2.5.1. *TCF3*::*HLF*

As a result of a translocation t(17;19)(q22;p13), the created *TCF3*::*HLF* fusion gene is a very rare abnormality (<0.7% of pediatric ALL cases) and highly resistant to standard treatment [94]. The *TCF3* (alias *E2A*) trans-activating domains are fused to the DNA-binding and dimerization domains of a hepatic leukemic factor in the *TCF3*::*HLF* fusion gene, resulting in a chimeric transcription factor (HLF). Such modifications cause mutarotation arrest in vivo, implying that transgenic mice’s susceptibility to infection was attributable to immunodeficiency. Furthermore, as the research shows, *TCF3*::*HLF* is involved in lymphoid development as well as leukemogenesis [95]. Simultaneously *TCF3* and *HLF* genes, and their rearrangements, also induce valid changes in patients with B-ALL. *TCF3* is a transcription factor that is required for B-cell lineage development. The 5′-end of the *TCF3* gene, which encodes transactivation domains, generally fuses with the 3′-DNA-parts of another gene, which encodes a particular DNA-binding domain, as in the *HLF* gene [96,97,98]. The *HLF* gene (locus 17q22) codes for a member of the proline- and acidic-rich (PAR) protein family, which is a subset of bZIP transcription factors. To trigger transcription, the encoded protein forms homodimers or heterodimers with other PAR family members and binds sequence-specific promoter sites [99]. Adel A. Hagag et al., in studies on a 44-patient cohort, suggested that PAR proteins are linked with poor prognosis [100]. It is important to distinguish between the partner genes for *TCF3*, as it is a critical issue for risk evaluation. 

The identification of BCL2 dependence in *TCF3*::*HLF* ALL as a druggable target demonstrates how the integration of drug response profiling and molecular genetic analysis might inform the development of novel treatment regimens in individuals with intractable therapeutic requirements. *TCF3*::*HLF* may have a transcriptional target in the BCL2 protein [101]. Resistance to apoptosis caused by large levels of the anti-apoptotic oncoprotein BCL2 may improve cancer cell survival and represent a druggable target [94]. One of BCL2’s targeted drugs is venetoclax, which is a highly selective inhibitor of the *BCL2* gene. The *BCL2* gene (locus 18q21.33) increases cell survival by blocking its programmed death [102]. Thus, proliferation leads to gaining resistance against used therapy. Venetoclax is not a gold standard against ALL with *TCF3*::*HLF* because there is a relapse of the disease over time [103]. This is associated with decreased venetoclax binding, increased anti-apoptotic family members associated with BCL-2, changes in the microenvironment, and malfunction of the TP53 pathway [103]. In the context of chronic venetoclax exposure, it increases *MCL1* and *BCL-XL* expression, both of which have antiapoptotic properties [104], leading to acquired resistance [103]. This suggests that drugs may lead to resistance by activating already existent cellular pathways.

Genetic mutations also influence other genes associated with resistance. The *MYC* gene and the expression of *MYC*-associated metabolic pathway genes play key roles in the escape of a malignant. *MYC* (locus 8q24.21) is a proto-oncogene that encodes a DNA-binding factor that may both stimulate and inhibit transcription. *MYC* controls the expression of several target genes, such as T*P53*, *TCF4*, and *E2F*, which regulate relevant biological activities—cell proliferation and cell cycle progression. *MYC* also plays an important function in DNA replication. Deregulation of *MYC* expression due to several types of genetic alterations [105] leads to constitutive *MYC* activity in various malignancies, including B-ALL, and promotes oncogenesis [106]. The chromosomal aberration *TCF3*::*HLF* stimulates *MYC* expression not only by its direct enhancer activation but also by inhibiting the ubiquitin ligase FBXW7. This ligase regulates MYC (and other targets) by ubiquitin-dependent protein degradation. *TCF3*::*HLF* thereby enforces a malignant cellular identity, allowing both self-renewal and cell proliferation [107]. Such co-occurrence and interdependence of gene influences in B-ALL are widely implicated in carcinogenesis and the acquisition of resistance to treatment by cancer cells. Table 5 summarizes mutations in genes that are responsible for a worse prognosis in patients with current *TCF3*::*HLF* translocation.

#### 2.5.2. t(9;22) Chromosome Philadelphia

Chromosome Philadelphia was discovered in 1960 by Peter C. Nowell and his student David Hungerford [108]. This mutation is based on the reciprocal translocation between chromosomes 9 and 22. The *Abelson murine leukemia* (*ABL*) gene (locus 9q34.12) encodes a cytoplasmic and nuclear protein tyrosine kinase that is responsible for increased proliferation, better viability, and migration changes. The whole process is under close regulation by hematopoietic growth factors. During the mutation, the ABL gene is translocated to band q11 on chromosome 22, where the *Break Point Cluster (BCR)* gene is located. *BCR* encodes proteins associated with kinase activity, and it is responsible for the oxidative burst in neutrophils. Each gene contributes its own domains [109]. In *ABL1*, we can distinguish the SH domain (SH1/SH2), Proline-rich domain, and DNA- and actin-binding domains. The domains from *BCR* include the CC domain, Ser/Thr kinase domain (contains docking site Y177), and Rho/GEF kinase domain. The CC domain and kinase domain are crucial for the activation of the *ABL1* gene. The SH domain of *ABL1* controls its activation and deactivation. Fusion between *ABL1* and *BCR* gene leads to the activation of Abl1 and Abl2 tyrosine kinase, which transmit signals by a number of pathways, such as the JAK/STAT pathway, mTOR pathway, MAPK/ERK pathway, TRIAL-included pathway, and CEBP-mediated differentiation [110,111]. Tyrosine kinase belongs to a group of enzymes that catalyze the transfer of a phosphate group from ATP to the target protein’s tyrosine residues. The kinase protein domains form the “cleft” that binds ATP. This activates signals that are transmitted from the surface to cytoplasmic proteins and the nucleus of the cell. The ABL1 and ABL2 kinases have an amino terminal “cap” region and a long carboxy tail. This special “cap” can bind lipids to stabilize inactive conformation and is required to achieve and maintain inhibition. The oncogenic ABL1 proteins lack that auto-inhibitory “cap” region and, in consequence, become an oncogene, which is characterized by uncontrolled proliferation, escape from the control of growth factors, and decreased apoptosis [112]. Despite this phenomenon, oncogenic tyrosine kinase can be inhibited with compounds called tyrosine kinase inhibitors (TKI). Most known TKI have one to three hydrogen bonds that bind with amino acids located in the “cleft” region. Therefore, they can compete with ATP and lead to the inhibition of the kinase. Tyrosine Kinase Inhibitors can be classified as type I or type II. Type I recognizes the active conformation of kinase; type II recognizes its inactive conformation [113]. 

The Philadelphia chromosome is detected in 5% of pediatric patients with ALL. Historically, this translocation was associated with poor prognosis and long-term event-free survival, reaching only about 30%. Autologous hematopoietic stem cell transplantation (HSCT) showed a better survival rate than chemotherapy, but only a small group of patients were qualified for it. In 2001, the Food and Drug Administration (FDA) approved imatinib for the treatment of adults with chronic myeloid leukemia (CML) [114]. In October 2002, a trial conducted by the Children’s Oncology Group began. A total of 93 patients with Ph+ participated in this study. Patients were divided into five cohorts, with longer exposure to imatinib in each one (cohort 1 = 42 days, cohort 5 = 280 days). The 3-year event-free survival for patients in the cohort with the longest exposure to imatinib was 80.5%. This was a spectacular result compared to the historical 30% [115]. Another European intergroup study of post-induction treatment of Philadelphia-chromosome-positive ALL (EsPhALL) started in January 2004. It included 179 patients, 44 of whom received imatinib. In this protocol, imatinib was added to standard chemotherapy after the induction phase. The 4-year disease-free survival was 72% [116]. The next EsPhALL study began in 2010 and included 155 patients. Imatinib was administered continuously along with chemotherapy from the 15th day of induction. The overall survival in this study was 71% [117]. In Poland, children with the presence of Ph (+) have been treated according to the EsPhALL2010 Protocol since 2012. The estimated 5-year survival rate for children is 74.1% [118]. 

Despite the great success and significant increase in the survival rate of Ph+ ALL patients, it is reported that about 25% of patients during treatment will develop intolerance or resistance to TKI [119]. The mechanisms of resistance to tyrosine kinase inhibitors can be divided into related or unrelated to mutations. Point mutations in *BCR::ABL1* genes are the most common mechanism of resistance. The first mutation detected in imatinib resistance cells was T315I, also known as the “gatekeeper mutation”. It causes a hydrogen bonding break, which is critical for drug binding [120]. This and many other point mutations (Table 2) have contributed to the development of second-generation TKIs—dasatinib and nilotinib. In 2014, the Chinese Children’s Cancer Group study started a clinical trial with 189 children. A total of 97 of them received imatinib and 92 received dasatinib. The second group of patients had significantly higher rates of 4-year event-free survival (71.0% vs. 48.9%), overall survival (88.4% vs. 69.2%), and lower relapse rates (19.8% vs. 34.4%) than the group treated with imatinib [14]. Dasatinib has the ability to bind the BCR::ABL1 tyrosine kinase, SRC family kinases, and c-Kit receptors [121]. Furthermore, dasatinib can cross the blood–brain barrier to eradicate central nervous system (CNS) blast cells. A study by Moshe Talpaz et al. suggested that less-stringent conformational requirements for BCR::ABL1 binding will be likely to make dasatinib active against kinase domain mutants responsible for imatinib resistance [122]. There have been more cases of recurrence or intolerance to imatinib/dasatinib. Therefore, another clinical trial was conducted on 15 patients with CML or ALL with such a disorder. Patients received nilotinib in cycles (from 12 to 24) for 28 days each. The study included four patients with ALL Ph+, and three of them achieved complete remission. Even though the number of patients is small, these results are very encouraging [123]. 

In Table 6, we present the point mutations of the *BCR*::*ABL1* gene that most often appear in cells with resistance to TKI [119,122,124,125].

As we can see, the T315I mutation and its varieties occur in the case of resistance to first- and second-generation TKI. This is a mutation resulting in the substitution of isoleucine for threonine, which makes the other generation inhibitors unable to bind and block the BCR-ABL1 kinase. We can come to the conclusion that T315I may be the biggest obstacle [126].

Besides the point mutation of the *BCR*::*ABL1* gene, a large overexpression of *BCR::ABL1* has been described that causes resistance by raising the quantity of the target protein that is to be inhibited by the drug’s therapeutic dosage. Thanks to this, blast cells can survive despite TKI presence [127].

Resistance mechanisms unrelated to *BCR::ABL1* mutations affecting all generations of TKI involve the activation of other pathways and changes in drug bioavailability. One of the described changes is the presence of the *MRD1* gene (locus 2q23.1). Its presence causes an increase in the *MRD1* gene product, P-glycoprotein (Pgp), an efflux pump, which is involved in “pumping” drugs out of the blast cells. This causes a reduction in drug concentrations and results in resistance to imatinib and nilotynib [120,127,128]. Another mechanism observed in cancer cells is the activation of other tyrosine kinases—for example, Lyn and Hck kinases. In their work, Mahon and colleagues suggest that a possible role of Lyn kinases is to increase the oncogenic potential of the BCR::ABL1 kinase [127]. 

In some cases, resistance can be overcome by increasing the dose. Unfortunately, tyrosine kinase inhibitors, apart from their anti-cancer effects, have strong side effects. Sometimes, these effects are so severe that they force a reduction in the dose or stop the drug administration completely. This is illustrated by an example of 31 patients treated in Polish Hemato-Oncology Centers, where the most common side effects were infections, hepatotoxicity, gastrotoxicity, dermatitis, and neuro and nephrotoxicity. The greatest range of these ailments, in grades from I to III, was observed after imatinib. A patient who received ponatinib developed dermatitis of grade IV [118]. 

A great promise for Ph+ patients is a third-generation drug—ponatinib. It interacts with the inactive *ABL* conformation, including cells with the highly resistant T315I mutation. In 2015, research by Jabbour was published, in which 37 patients received ponatinib. Patients received TKI during induction and maintenance. After 2 years of follow-up, the overall survival was 80% [129]. However, J. Chen et al. reported three cases of resistance during therapy with ponatinib. These cases were a 29-year-old female patient with CML, a 46-year-old female patient with ALL, and a 7-year-old male patient with ALL. The latter’s treatment started with imatinib, during which various mutations were identified—D276G, F311I, and F317L. After 18 months, imatinib was replaced with ponatinib. Then, after 5 months, new mutations appeared—F311I/T315I and D276G/T315L. After consecutive months of chemotherapy, only the D276G/T315L mutation was found. Since D276G is known to be sensitive to TKI, it is obvious that the T315L mutation was responsible for ponatinib resistance [130]. Table 7 shows the characteristics of genes present in ALL Ph+ patients responsible for treatment resistance.

### 2.6. ALL Ph-like Subtype, Active Signal Pathways, and Resulting Treatment Opportunities

Philadelphia-chromosome-like (Ph-like), otherwise known as BCR-ABL1-like ALL, is a distinct subtype of high-risk leukemia. It is characterized by having a gene expression signature similar to Ph-positive ALL but lacks the BCR-ABL gene fusion. The mutations found in this leukemia affect genes involved in B-lineage differentiation and maturation, regulation of B-lymphocyte proliferation, and activation of metabolic pathway kinases that promote leukemogenesis. In pediatric patients, the frequency of Ph-like ALL is up to 15% [131], the risk of relapse is 50%, and the 5-year survival is estimated at about 62% [132]. Ph-like ALL is an extremely heterogeneous group, and diagnosis can only be made by identifying relevant abnormalities in the genome of leukemic cells. Anomalies affecting different genes, such as rearrangements, sequence mutations, and copy number changes, allow us to distinguish four main groups. By activating different metabolic pathways, the possibility of introducing targeted treatment is achieved. 

The first group is associated with the activation of the JAK/STAT metabolic pathway. Janus family enzymes (JAKs) are tyrosine kinases that are involved in cytokine receptor signaling in hematopoietic cells. Their abnormal function promotes uncontrolled proliferation and the growth of leukemic cells [133]. 

The most common cause of JAK/STAT pathway activation is point mutations in the JAK2 domain [134]. These occur in cells with present rearrangements and mutations, most commonly in *CRLF2* and *IKZF1* genes. Other rearrangements in which the activation of this pathway is observed include JAK2, EPOR, TYK2, IL7R, and SH2B3. Gene *CRLF2* (locus Xp22.33) encodes a receptor for thymic stromal lymphopoietin that participates in the activation of the STAT transcription factor through the JAK pathway [135]. The *IKZF1* gene (locus 7p12.2) encodes a zinc-finger transcription factor involved in lymphoid development. Defects in this factor result in the expression of specific genes, which ultimately leads to the development of leukemic cells [136].

In this group, the targeted treatment—a JAK/STAT pathway inhibitor—can be added to standard chemotherapy. Ruxolitinib is an FDA-approved JAK inhibitor that binds to active *JAK2* (locus 9p24.1) at the ATP binding site. In the St. Jude Total XV study, which enrolled patients with B-cell ALL and T-cell ALL, a group of patients with an activated JAK/STAT received ruxolitinib after the induction phase for 28 to 42 days. The overall survival (for all of the patients in this study) was 94% [137]. In clinical studies conducted in vitro by Charlotte E. J. Downes and colleagues, three mutations were found that are responsible for ruxolitinib resistance. They relate to the *JAK2* kinase domain, including *JAK2* p.Y931C, *JAK2* p.L983F, and *JAK2* p.G993A. It is suggested that a patient treated with ruxolitinib in monotherapy may develop such mutations within three months. It is also disturbing that in such patients, the blast cells may acquire cross-resistance to TKI and JAK inhibitors [138]. To date, only one case of ruxolitinib resistance in a high-risk pediatric ALL patient has been reported. In a patient who relapsed on treatment after 12 months, the *JAK2* R938Q mutation was found. This leads to a substitution in the C-terminal kinase domain of JAK2 near the adenosine triphosphate (ATP) loop and substrate binding site. This change is likely to result in a loss of the inhibitory site for ruxolitinib [139]. 

Recently, Shirazi et al. conducted an in vitro study on cells with the presence of *MYB*::*TYK2* fusion with an active JAK/STAT pathway. In this study, the researchers examined how cancer cells behave after exposure to the JAK/STAT pathway inhibitor—cerdulatinib. The leukemic cells were exposed to cerdulatinib for 151 days. Prolonged exposure to the drug induced an increase in JAK/STAT signaling, JAK1 overexpression, and subsequent development of drug resistance. At the same time, this resistance was shown to be reversible, as after the withdrawal of the drug, cells became sensitive to it again [140].

Second-class rearrangements are related to the fusion of *ABL* genes, which includes *ABL1*, *ABL2*, *PDGFRA*, *PDGFRB*, and *CSF1R*. Each of these genes is a proto-oncogene that encodes a protein involved in many cellular processes, including cell division and differentiation, specifically tyrosine kinase and the JAK/STAT pathway [109,141,142]. The proliferation of leukemic cells with an “abl-class” fusion presence can be effectively inhibited by TKIs. Schwab et al. conducted a study on 15 patients with the presence of *EBF1*::*PDGFRB* fusion. In two patients with a poor response, imatinib was added to standard chemotherapy. One of these patients achieved remission at the 14th week of treatment with imatinib. The second patient died, but as a result of post-transplant complications [143]. Another study by Tanasi et al. included 24 patients with pediatric ALL with *ABL1* class fusion who were treated with a combination of TKIs and chemotherapy. A total of 19 patients were treated with imatinib during frontline therapy; one of them had to be switched to dasatinib. In 78% of patients, within 2.5 months of exposure to TKI, MRD was less than 10^−4^. These results show promising MRD responses and outcomes [144]. 

The third group includes mutations that activate the RAS metabolic pathway (NRAS, KRAS, PTPN11). Der-Cherng Liang et al., in a clinical trial involving 535 children, showed that NRAS mutations were associated with a higher frequency of hyperdiploidy and a lower frequency of *ETV6*::*RUNX1*, while *KRAS* mutations were associated with younger age, a higher frequency of KMT2A rearrangements, but no significant difference in event-free survival [145]. 

The last, fourth class involves less-common fusion, such as *FLT3*, *FGFR1*, and *NTRK3* [146]. One of the rare ones, occurring in only about 1% of patients, is *ETV6*::*NTRK3* fusion. *ETV6* (locus 12p13.2) [147] belongs to the erythroblast-transformation-specific family and is a transcriptional corepressor in hematopoiesis [148]. The *NTRK3* gene (locus 15q25.3) encodes TRKC, a member of the tropomyosin receptor tyrosine kinase (TRK) family [149,150]. Activation of the TRK pathway has been described in a variety of cancers, including B-ALL. In vitro studies were performed on mouse leukemic cells with *ETV6*::*NTRK3* fusion present. The cells were treated with a TRK inhibitor—larotrectinib. After monotherapy lasting 6 weeks, a significant reduction was achieved. Cytometric examination revealed the absence of blasts in the cell marrow [151]. Drilon and colleagues conducted a study on 55 patients with current TRK fusion. The patients were of different ages (less than 2 years to more than 40 years) and with different malignancies. Patients received larotrectinib orally for the duration of treatment; the overall response rate was 80% [152]. These results show that treatment may be effective in children with ALL, but clinical trials must be carried out. Table 8 shows the characteristics of genes of which mutation leads to activation of metabolic pathways in ALL Phlike patients.

Due to the huge number of mutations that may appear in this subtype and the identification of changes in cells in vitro and in vivo, it is predicted that resistance in children with Ph-like ALL will continue to develop.

### 2.7. Monoclonal Antibodies, CAR-T Cell Therapy—Has There Been Any Resistance to the Newest Treatment Methods?

Immunotherapies specific for B-cell precursor acute lymphoblastic leukemia, such as anti-CD19 chimeric antigen receptor (CAR) T-cells and blinatumomab, have dramatically improved the therapeutic outcome in refractory cases. The European Medicines Agency approved blinatumomab for ALL patients in 2015 [153]. Although ozogamicin inotuzumab (InO) has been shown to be effective in adults, due to little information on its safety and efficacy in children, it is not a registered drug for patients under 18 years old. Nevertheless, children receive InO under the compassionate use program. Immunotherapies focus on surface antigens present on blast cells. CD19, CD20, or CD22 antigens are viable goals for targeted therapy in patients with B-ALL. 

Blinatumomab is a CD3/CD19-bispecific (BiTE) antibody. It is composed of two single chains that are binding spots for the CD3 antigen of T lymphocytes and the CD19 antigen present in cancer cells [154]. The CD19 antigen has four splice variants: a full-length transcript, a transcript with exon 2 omitted, a transcript with part of exon 2 omitted, and a transcript without exons 5–6. Blinatumomab binds to the CD19 and CD3 antigens. This creates a kind of synapse that allows the activation of T lymphocytes. The release of cytokines such as IL-1, IL-6, INF, and tumor necrosis factor causes the death of blast cells (Figure 1 a). However, one of the main side effects that limit the use of blinatumomab is Cytokine Release Syndrome (CRS) and neurological disorders [153]. The Blinatumomab effect has potential, but research is still needed to make it close to 100% effective, excluding resistance mechanisms. In their work, Orlando and colleagues showed the presence of mutations in exons 2 to 5 of the CD19 antigen. The presence of an anti-CD19-antibody (FMC63)-binding site on exon 4 implies a mutation in the splice-site acceptor (SSA), which leads to intron retention and transmembrane domain instability. Mutations in these exons result in damaged protein in the transmembrane domain. As a result of this, the CD19 antigen is unable to properly adhere to the cell surface [155]. Deng et al. showed that malignant B cells that express CD19 transcripts without exon 2 had reduced *SRS3* gene expression [156]. The *SRS3* gene (locus 6p21.31-p21.2) encodes a splicing factor that affects pre-mRNA processing and is involved in the recognition of terminal exons [157,158]. Therefore, the *SRS3* gene has been implicated as a splicing factor that causes CD19 transcription without exon 2. 

Another mechanism was investigated by Braig and colleagues. Neoplastic cells from relapsed patients were tested for the presence of the CD81 antigen. This is a protein that assists with the maturation of the CD19 antigen in the Golgi apparatus. CD81 also allows antigen CD19 to be safely “transported” from the Golgi apparatus to the cell surface. In the absence of CD81, these processes are unable to occur, so the leukemic cells again lose the CD19 antigen on their surface [159]. As a result of these mutations, blast cells are missing the CD19 antigen, and at the same time, lose the attachment point for Blinatumomab (Figure 1b). 

An antibody that is used in highly resistant patients is Inotuzumab ozogamicin (InO). The FDA approved inotuzumab ozogamicin (BESPONSA) for the treatment of adults with relapsed or refractory B-cell precursor acute lymphoblastic leukemia on 17 August 2017, but not for children, where studies are ongoing [160]. InO is a humanized anti-CD22 monoclonal antibody–drug combination. It binds with high affinity to CD22, a cell surface antigen that is expressed by over 90% of B-cell blasts in almost all patients with B-cell ALL. Metabolism of the drug causes the release of intracellularly unconjugated calicheamicin responsible for DNA binding and cleavage, which in turn leads to the apoptosis of a blast cell (Figure 1c) [161,162]. Hagop M. Kantarjian et al. suggest that InO can be utilized as an effective bridge to hematopoietic stem cell transplantation, and it is related to improved overall survival and progression-free survival in patients with recurrence/relapse B-ALL when compared to standard-of-care (SoC) chemotherapy [161]. In Deepa Bhojwani et al.’s study, patients treated with Besponsa with overt relapse, complete remission was achieved in 28 of 42 (67%) cases. Researchers did not find a meaningful predictive signal for response in any baseline patient or illness characteristic (age, sex, cytogenetic subtype) [162]. In a case presentation with InO usage presented by Megan R. Paul et al., CD22 antigen loss was observed (Figure 1d). A cytogenetic study demonstrated the persistence of *ETV6*-*RUNX1* fusion, deletion of 12p13, and trisomy 10. Multiple structural and numerical chromosomal abnormalities were also found in the blast population, including reciprocal translocation between 5p10 and 10q10, reciprocal translocation between 11p21 and 12q24.1, and deletion from 15q15 to 15q26.1. On chromosome 19, the *CD22* gene was expressed. The cytogenetic report revealed no chromosome 19 abnormalities [163]. 

Therapies with InO and Blinatumomab are still being studied and developed. The latest figures show that about ten percent of patients develop a CD19-negative relapse after treatment with Blinatumomab [164]. The use of Besponsa in pediatric patients still requires many studies.

CAR-T (Chimeric Antigen Receptors—CARs) cell therapy is a relatively new and promising ALL treatment approach that works by affecting the antigens of cells. In CAR-T therapy, as in classical adoptive therapy, lymphocytes are collected from a patient, subjected to tumor-specific genetic modification and expansion in in vitro cultures, and then administered intravenously to the patient. Genetic modifications, in this case, refer to genes encoding chimeric antigen receptors. Activation of T lymphocytes is determined by two signals; the first is detected by a T-cell receptor (TCR), and the second engages surface antigens, especially CD20/CD80 or 4-1BB/4-1BBL [165]. These receptors enable the specific recognition of cancer cells and, when bound to their surface, induce lymphocyte cytotoxicity, which leads to the killing of the cancer cell. Therapies using genetically modified lymphocytes have shown very high efficacy in clinical trials, especially in hematologic malignancies [166,167,168]. Every phase of CAR-T creation performed in vitro is significant for maintaining the construct’s appropriate architectural features and, as a result, affects the CART mode of action in vivo. Clinical trials in hemato-oncology yielded promising results for CART’s efficacy in ALL. Primary T cells were engineered in 1992 and developed rapidly. In August 2017, the American Food and Drug Administration approved Car T-Cell therapy for patients under 25 years old with the relapse of or refractory ALL [169]. In Poland, the first CAR-T cell therapy was carried out in 2019 [170]. According to many clinical trials, the remission rate has increased from 70% to 90% [154,171,172]. 

However, there is a risk that some patients will acquire resistance to CAR-T therapy [166]. This might be a consequence of the continued exposure to modified cells, especially in patients with relapsed ALL who have a negative expression of CD19 or a CD19 splice variant lacking exon 2 that is recognized by CAR-T scFv cells [173]. Following studies revealed that some CAR-T escape, causing changed CD19 isoforms were not created during therapy but were present at the time of diagnosis. They can evolve and become a significant clone of cancer cells. Sotillo et al. came to the same result. In a group of 14 children with CD19+ B-ALL, they looked at the expression of CD19 isoforms. Three distinct bands were obtained based on bone marrow and peripheral blood samples, semiquantitative CD19 cDNA amplification by RT-PCR, and visualization by agarose gel electrophoresis. These were the same size as full-length CD19 (800 bp) and exon 2 defective isoforms found following CAR-T therapy [174]. This allows for the identification of previously identified extracellular epitopes (which are alternatively spliced in leukemia compared to normal B cells), paving the way for the development of new CAR-Ts that target alternative CD19 ectodomains. As a result, treating many epitopes with combination therapy may increase survival in patients with treatment-resistant B-ALL [175].

Antigen loss or down-regulation is another type of treatment resistance, well known for antibody therapies. In rare cases, disease relapse results in cancer cells no longer expressing the antigen targeted by CAR-T therapy (CD19 and/or CD22) [176]. Alternative splicing, which yields CD19 isoforms with disruption of the target epitope and/or reduced cell surface expression, is one of the established pathways leading to the loss of antigen CD19 [174,177]. With anti-CD22 CAR-T cell therapy, a simple quantitative reduction in CD22 expression or density in the leukemic population was sufficient to avoid CAR-T cells, thereby allowing leukemia relapse despite persistent CD22 positivity, with individual differences in threshold antigen density that were responsible for relapse or resistance [178]. Another limitation for CAR-T-cell therapy is its side effects, which (similar to Blinatumomab) are neurotoxicity and Cytokine Release Syndrome (CRS). Patients with the most severe reactions have been shown to express high levels of interleukin IL-6. CRS can be relieved by infusion of the monoclonal antibody tocilizumab or siltuximab, which blocks the IL-6 receptor and by that reduces inflammation [179,180]. CRS occurs in almost every patient within 1–21 days of T-cell infusion; its severity depends on CAR-T cell expansion and immune activation. Neurotoxicity can be secondary to CRS and, with the proper treatment, can be reversible [165,181]. 

The genetic changes manifested by resistance to CART treatment involve more than molecules contained on cancer cells’ surfaces. In Atsushi Watanabe et al.’s study, they analyzed TNF-related apoptosis-inducing ligand (TRAIL) resistance and epigenetic modification of death receptor genes in childhood pre-B-ALL. TRAIL is a cytotoxic agent that promotes graft-versus-leukemia (GVL) and induces apoptosis in target leukemia cells via its death receptors (DR4 and DR5). The GVL effect is an immunologically significant phenomenon that lowers the rate of leukemia relapse following allogeneic bone marrow transplantation [182]. In turn, *DR4* (locus 8p21.3) [183] and *DR5* (locus 8p21.3) [184] genes facilitate the selective elimination of malignant cells through the induction of apoptosis [185]. In a study where 459 patients’ samples were diagnosed, Watanabe looked at TRAIL resistance caused by hypermethylation of the *DR4* and *DR5* genes and concluded that it was improbable, especially in cases where favorable karyotypes such as hyperdiploidy and *ETV6-RUNX1* were present. Their findings give an epigenetic basis for immunotherapy success in BCP-ALL patients, as the TRAIL/death receptor system plays an important role in the anti-leukemic actions of anti-CD19 CAR-T cells. Furthermore, in some BCP-ALL cases with unfavorable karyotypes, such as dic(9;20), *KMT2A*-rearranged, and hypodiploidy, determining the methylation status of the *DR4* and *DR5* genes could be clinically useful in predicting immunotherapy success [186]. Table 9 shows genes and mutations characteristics that cause resistance to new targeted medicines.

CAR-T cell therapy has brought with it high hopes for patients who are not responding to currently available therapies. Although it significantly increases cure rates, the advanced, highly refractory disease can lead to multiple organ failure and eventually death of the patient before receiving the therapy. 

## 3. Conclusions and Future Perspective

Due to enormous advances in medicine, patients with ALL have a good chance to achieve remission. Genetic profiling studies allowed us to identify ALL subgroups with different cure rates. Given the heterogeneity of these groups, treatment appropriate to the mutation or active signaling pathway is needed. It is difficult to comprehend that, despite such knowledge about blast cell biology and a wide range of therapeutic options, still up to 30% of patients do not achieve complete remission. Another troubling phenomenon is the lack of response to targeted treatment—using antibodies or CAR-T cells therapy. As we have outlined, a cancer cell has enormous potential to escape from treatment (Figure 2). Additionally, it is constantly able to improve itself and develop new ways to do so. Until we know how to effectively achieve remission in the great majority of patients, research on ALL will continue.

## Figures and Tables

**Figure 1 ijms-23-03067-f001:**
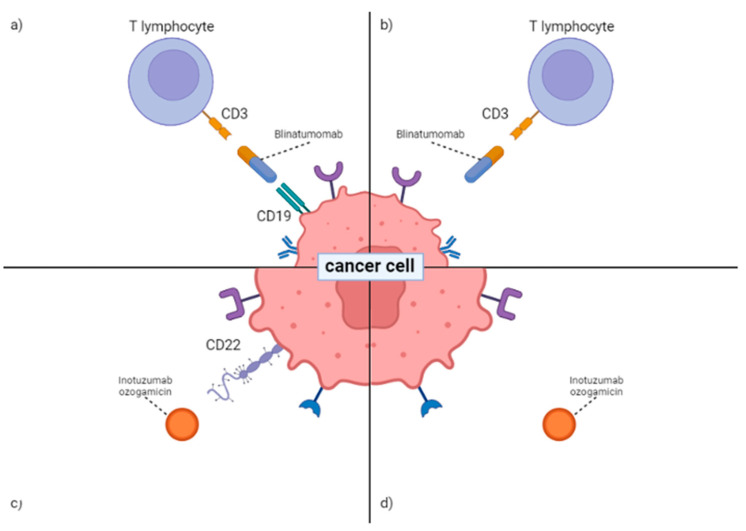
Effect of Blinatumomab and Inotuzumab ozogamicin (InO) on a cancer cell and the mechanism of tumor cell escape. (**a**) Blinatumomab in therapeutic context. Blinatumomab is a mediator between CD19 antigen on T lymphocytes and CD19 antigen localized on cancer cells. Effect of connection is cancer cell death, caused by cytotoxic activity of T lymphocyte; (**b**) Resistance to Blinatumomab. No CD19 antigen on cancer cells leads to no connection between T lymphocyte and cancer cell, where there is no cell death; (**c**) Inotuzumab ozogamicin (InO) in therapeutic context. When CD22 antigens are found on cancer cells, InO binds, and it causes the production of intracellularly unconjugated calicheamicin, which causes blast cell death. (**d**) Resistance to InO. InO has no effect when CD22 antigens are not present on cancer cells. Image created with BioRender.com, accessed on 29 January 2022.

**Figure 2 ijms-23-03067-f002:**
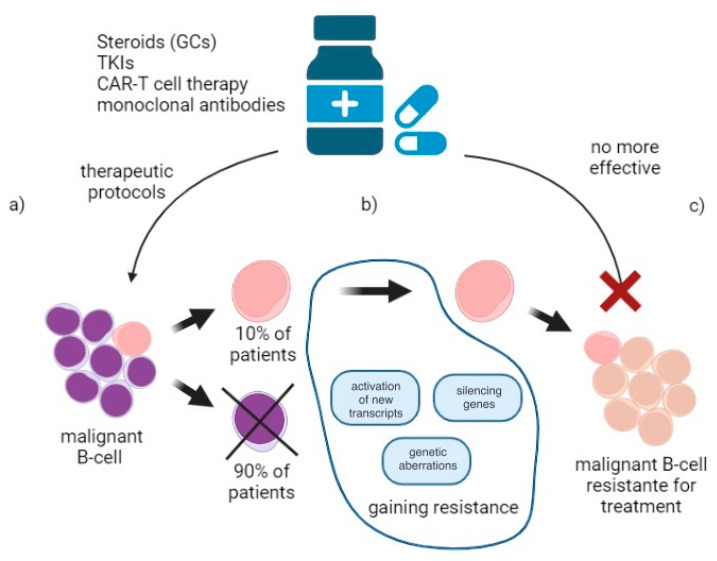
Response to B-ALL treatment. (**a**) Effective influence of therapeutic protocols. Patients are curable in 90% of cases, which still leaves 10% og patients resistant to therapeutic protocols; (**b**) Overcoming treatment. Cancer cells gain resistance by multiple changes in construction and activeness, such as activation of new transcripts, silencing genes, gene aberration, gene impact for expression of genes—changes observable in metabolic cell’s proteins; (**c**) Resistance to treatment. Treatment is overcome by cancer cells through genetic modifications, which protect it from the typical protocol treatment.

**Table 1 ijms-23-03067-t001:** Selected differences between protocols and the comparison of their cure rates.

Protocol	Genetic Anomalies Characteristic for HR	Drugs Used in Induction	Drugs Used in Consolidation	Drugs Used in Intensification	Maintenance of Remission—Drugs Used and Duration	Radiotherapy	Curability	Reference
AIEOP BFM ALL 2017	*KMT2A-AFF1* *TCF3-HLF* hypodiploidy not ETV6-RUNX1 + any KMT2A rearrangement	Prednisone VCR DNR Oncaspar MTX i.t.	Dexamethasone ARA-C VCR Oncaspar CMP 6-MP MTX i.t.	Dexamethasone VCR DOXO Oncaspar CMP TG MTX i.t.	MTX p.o. 6-MP MTX i.t (every 6 weeks for HR patients) 74 weeks for boys and girls	Only for patient with CNS3 status; older than 4 years old	95%	[10,11]
UK ALL 2011	iAMP21 t(17;19) q(22;p13)//TCF3(E2A)-HLF MLL rearrangement near haploidy hypodiploidy	Dexamethasone VCR Oncaspar 6-MP MTX i.t.	6-MP MTX i.t. CMP ARA-C VCR Oncaspar	Dexamethasone VCR DOXO Oncaspar MTX i.t. CMP ARA-C 6-MP	4 regimens of maintenance depending on the risk group; dexamethasone VCR 6-MP MTX p.o. MTX i.t. 2 years for girls 3 years for boys	Only for patient with CNS3 status	91.5%	[12,13]
CCG-ALL-2015	t (1;19), t (9;22), MLL rearrangement hypodiploidy iAMP21 BCR-ABL fusion E2A-PBX1 fusion	Dexamethasone (day 1–4) prednisone (day 5–28) VCR DBR Oncaspar CMP ARA-C 6-MP Triple IT = MTX + H-C + ARA-C	HD-MTX 6-MP	Dexamethasone VCR DNR ARA-C Oncaspar Triple IT	6-MP + MTX p.o. + Triple IT every 4 or 7 weeks: dexamethasone VCR CMP ARA-C 76 weeks for boys and girls	Only for patient with CNS3 status; older than 3 years old	90%	[14,15]
JACLS	t(4;11) or t(1;19) KMT2A/AFF1 hypodiploidy TCF3/PBX1	prednisone (day 1–7) Dexamethasone (day 8–14) prednisone (day 15–28) VCR CMP THP-adriamycin Oncaspar MTX i.t. on day 1 Triple IT on day 8.2	CMP ARA-C THP-adriamycin 6-MP Triple IT	Prednisone VCR THP-adriamycin CMP Oncaspar Triple IT	98 weeks; divided into 4 stages 1A—MTX p.o + 6-MP + Triple IT 1B—MTX p.o. + 6-MP + Triple IT + radiotherapy 2—prednisone + VCR + CMP + Oncaspar 3—MTX p.o + 6-MP + Triple IT 4—prednisone + VCR + THP-adriamycin + Oncaspar	During maintenance 1B for CNS-positive status	96.4%	[16]
COG-AALL	BCR-ABL fusion transcript t(9;22)(q34;q11) hypodiploid MLL rearrangement	Dexamethasone VCR Oncaspar MTX i.t. ARA-C i.t. extended induction: DNR	Dexamethasone VCR 6-MP MTX p.o. MTX i.t.	Dexamethasone VCR DOXO CMP ARA-C Oncaspar TG MTX i.t.	Dexamethasone 6-MP MTX p.o. VCR MTX i.t. on day 1 2 years for girls 3 years for boys	Only for patients with CNS3 status	95%	[17]

Comparison of protocols used in ALL treatment. AIEOP BFM ALL 2017—International collaborative treatment protocol for children and adolescents with acute lymphoblastic leukemia, UK ALL 2011—United Kingdom National Randomised Trial For Children and Young Adults with Acute Lymphoblastic Leukaemia and Lymphoma 2011, CCG-ALL-2015—Chinese Children Cancer Group Study, JACLS—Japan Association of Childhood Leukemia Study, COG-AALL—Children Oncology Group Protocol, VCR—Vincristine, DNR—Daunorubicin, DOXO—Doxorubicin, MTX—Methotrexate, ARA-C—Cytarabine, CMP—Cyclophosphamide, TG—Tioguanina, 6-MP—6-Mercaptopurine, H-C—Hydrocortisone, CNS—Central Nervous System p.o.—per os, i.t.—intrathecal.

**Table 2 ijms-23-03067-t002:** Characteristics of genes, their mutations, and pathological and treatment effects responsible for steroid, MTX, and asparaginase resistance.

Gene and Locus	Gene’s Mutation	Effects on Pathology	Effects on Treatment
*NR3C1* (locus 5q31.3)	Point mutations	Disordered transmission of signals via the NR3C1 receptor	Resistance to prednisone and dexamethasone
*CREBBP* (locus 16p13.3)	Sequence, deletion, missense mutations	amino acid substitutions in the histone acetyltransferase (HAT) domain	Resistance to steroids
*DHFR* (locus 5q14.1)	Single amino acid changes in DHFR, increased DHFR overexpression, decreased affinity of MTX for DHFR	Changes in metabolism of MTX	Resistance to MTX
*TYMS* (locus 18p11.32)	High expression of *TYMS*	Enhances purine production, cell proliferation	Resistance to MTX
*FPGS* (locus 9q34.11)	Low expression of *FPGS*	Plays role in MTX long-chain creation	Resistance to MTX
*RFC1* (locus 4p14)	Less expression of *RFC*	insensitivity to apoptosis mechanisms	Resistance to MTX
*ASNS* (locus 7q21.3)	High expression of *ASNS*	Excessive synthesis of asparagine	Resistance to asparaginase

**Table 3 ijms-23-03067-t003:** Characteristics of genes, their mutations, and pathological and therapeutic effects responsible for resistance to steroids and purine analogs.

Gene and Locus	Gene Mutation	Effects on Pathology	Effects on Treatment
*NT5C2* (locus 10q24.32-q24.33)	High expression	Controls the number of purine nucleotides in the cell and their outflow	Resistance to 6-MP
*PRSP1* (locus Xq22.2), *PRSP2* (locus Xq22.3)	Low expression	Purine and pyrimidine biosynthesis disorder	Resistance to 6-MP and 6-TG
*MSH6* (locu 2p16.3)	Hemizygous deletion, downregulation	Despite the presence of thiopurines, the loss of MSH6 produces a lack of cell apoptosis, which leads to an increase in blast survival	Resistance to 6-MP and prednisone
*SETD2* (locus 3p21.3)	Loss-of-function, frameshift, nonsense mutations, low expression	Tumor suppression is absent	Resistance to 6-MP

**Table 4 ijms-23-03067-t004:** Characteristics of genes occurring together with hyperdiploidy responsible for resistance to applied treatment.

Gene and Locus	Gene Mutation	Effects on Pathology	Effects on Treatment
*KRAS* (locus 1p13.2)	High expression	Attracts and activates growth factor-related proteins, as well as c-Raf and PI 3-kinase.	Increased resistance to MTX, increased sensitivity to vincristine
*FLT3* (locus 13q12.2)	Tandem duplication, high expression	Ras/Raf/MAPK pathway activation, misregulation of cellular activities, growth of B-Cells	Resistance to RTK and FLT3 inhibitors
*DOTL1* (locus 19p13.3)	High expression	Ras/Raf/MAPK pathway activation, misregulation of gene transcription, development, cell cycle progression, somatic reprogramming, and DNA damage repair	Resistance to RTK inhibitors
*PTPN11* (locus 12q24.13)	High expression	Increases SHP2 protein production, which leads to activation of Ras/Raf/MAPK pathway	Resistance to PTP inhibitors

**Table 5 ijms-23-03067-t005:** Characteristics of genes, mutations, and their effect on treatment present in patients with a *TCF3*::*HLF* translocation.

Gene and Locus	Gene Mutation	Effects on Pathology	Effects on Treatment
*TCF3* (locus 17q22)	Fusion with *HLF*	Drives lineage identity and self-renewal by recruiting *HLF* binding sites to hematopoietic stem cell/myeloid lineage-associated enhancers	Resistance to cell apoptosis and venetoclax
*BCL2* (locus 18q21.33)	High expression	Prolongs cell life by preventing the cell’s scheduled demise	Resistance to cell apoptosis and venetoclax
*MYC* (locus 8q24.21)	Deregulation of *MYC* expression	Has constitutive *MYC* activity and promotes oncogenesis	Resistance to treatment

**Table 6 ijms-23-03067-t006:** Point mutation of *BCR*::*ABL1* gene found in cells with resistance to imatinib, dasatinib, and nilotinib.

Imatinib	Dasatinib	Nilotinib
L248R E255V T315I T315V G250E Y253H E255K F359I	L248R T315I T315A T315V F317R F317V	L248R Y253H E255V T315I T315V F359I L248R G250E

**Table 7 ijms-23-03067-t007:** Characteristics of genes associated with treatment resistance in patients with the Philadelphia chromosome.

Gene and Locus	Gene’s Mutation	Effects on Pathology	Effects on Treatment
*BCR*::*ABL1* fusion (locus 9q34::22q11)	Point mutations	Hydrogen bonding break—critical for drug binding	Resistant to first and second generations of TKIs
*MRD1* (locus 2q23.1)	Presence in blast cells	Increased drug efflux	Resistance to TKIs of all generations

**Table 8 ijms-23-03067-t008:** Summary of genes responsible for activating metabolic pathways in ALL Ph-like; their mutations and treatment effects.

Gene and Locus	Gene Mutation	Effects on Pathology	Effects on Treatment
*CRLF2* (locus Xp22.33)	Rearrangements and point mutations	JAK/STAT pathway activation	Use of JAK/STAT pathway inhibitor
*IKZF1* (locus 7p12.2)	Rearrangements and point mutations	JAK/STAT pathway activation	Use of a JAK/STAT pathway inhibitor
NTRK3 (locus 15q25.3)	Fusion with ETV6	Activation of the TRK pathway	Use of a TRK inhibitor

**Table 9 ijms-23-03067-t009:** Characteristics of genes and their mutations responsible for resistance to new targeted therapies.

Gene and Locus	Gene’s Mutation	Effects on Pathology	Effects on Treatment
*SRS3* (locus 6p21.31-p21.2)	Rearrangements, point mutations, decreased expression	Misregulation of splicing factor leads to lack of recognition of terminal exons	Resistance to blinatumomab
*DR4* (locus 8p21.3) *DR5* (locus 8p21.3)	Rearrangements, point mutations, decreased expression	Absence of apoptosis-induced selective apoptosis to eliminate cancerous cells	Increase in survival rate, frequent recurrences

## Data Availability

Not applicable.

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
