# Peer review of "Resistance Mechanisms in Pediatric B-Cell Acute Lymphoblastic Leukemia"

_ijms, 2022, doi:10.3390/ijms23063067_

Round 1

Reviewer 1 Report

In this work the group of Lejman provides a well written and exhaustive review exploring a wide range of mechanisms involved in therapy resistance in B-ALL in pediatrics population.  

They also provide a broad overview in the use of monoclonal antibodies e new CART cell therapy.

The concluding paragraph concerns new therapeutic strategies exploiting the processes characterizing the advent of Leukemia Initiating Cells.

Minor comments:

- The authors should proofread the review, as there are a few minor typographical and grammar errors.

- Please if mutations are specified with amino acids, they should not be written in italics

-  Chapters and subchapters are well written but very long and complex. I would suggest, if possible, to add at the end of them a table to summarize, for instance, genes or gene mutations’ effects on the pathology, or mechanism, or on the several treatments.

Kind regards

Author Response

Response to Reviewer 1 Comments:

Dear Sir or Madam, thank you very much for the review of our manuscript entitled: Resistance mechanisms in pediatric B-cell acute lymphoblastic leukaemia.

In response to your comment, we would like to thank you for appreciating our manuscript.

Comment 1.

The authors should proofread the review, as there are a few minor typographical and grammar errors.

Revision and our comments:

The errors were corrected.

Comment 2.

Please if mutations are specified with amino acids, they should not be written in italics

Revision and our comments:

The names of mutations were corrected.

Comment 3.

Chapters and subchapters are well written but very long and complex. I would suggest, if possible, to add at the end of them a table to summarize, for instance, genes or gene mutations’ effects on the pathology, or mechanism, or on the several treatments.

Revision and our comments:

We added additional tables and were citated in the text.

Best regards 

Monika Lejman

Reviewer 2 Report

This is an excellent review of current treatment protocols and resistance mechanisms in pediatric ALLs. The topic is extensively covered, and all recent relevant references are included. The manuscript is well structured and is pleasant to read. Significant effort from the authors. There are some language style issues with writing, which I believe will be taken care of in the proofing stage. No suggestions from my side. Thank you for the opportunity to review this manuscript.  

Author Response

Dear Sir or Madam, thank you very much for the review our manuscript

Thank you for your great comments.

Best regards

Monika Lejman

Reviewer 3 Report

In this outstanding manuscript, JÄ™draszek1 et al review current therapies in B-Acute Lymphoblastic Leukemia (B-ALL). The text is very informative, reviews most evidence in basic and clinics and presents different treatments (from current protocols, to glucocorticoids and cytostatics). It is very interesting how the authors review the molecular mechanisms and mutations driving sensitivity or resistance to treatments. 

I have no further comments and would like to thank the authors for this piece of work. 

Author Response

Dear Sir or Madam, thank you very much for the review our manuscript.   Thank you very much for your great comments.   Best regards   Monika Lejman